# Bakkenolides and Caffeoylquinic Acids from the Aerial Portion of *Petasites japonicus* and Their Bacterial Neuraminidase Inhibition Ability

**DOI:** 10.3390/biom10060888

**Published:** 2020-06-10

**Authors:** Hyun Sim Woo, Kyung-Chul Shin, Jeong Yoon Kim, Yeong-Su Kim, Young Jun Ban, Yu Jin Oh, Hae Jin Cho, Deok-Kun Oh, Dae Wook Kim

**Affiliations:** 1Plant Resource Industry Division, Forest Plant Industry Department, Baekdudaegan National Arboretum, Bonghwa-gun 26209, Korea; whs0428@bdna.or.kr (H.S.W.); yskim@bdna.or.kr (Y.-S.K.); oyj0705@bdna.or.kr (Y.J.O.); hjlife@bdna.or.kr (H.J.C.); 2Department of Bioscience and Biotechnology, Konkuk University, Gwangjin-gu, Seoul 05029, Korea; hidex2@naver.com (K.-C.S.); deokkun@konkuk.ac.kr (D.-K.O.); 3Division of Applied Life Science, Gyeongsang National University, Jinju 52828, Korea; yoon24@gnu.ac.kr (J.Y.K.); banyoung972@naver.com (Y.J.B.)

**Keywords:** bacterial neuraminidase inhibitors, bakkenolides, caffeoylquinic acid, *Petasites japonicus* plant extract, competitive inhibition, non-competitive inhibition

## Abstract

*Petasites japonicus* have been used since a long time in folk medicine to treat diseases including plague, pestilential fever, allergy, and inflammation in East Asia and European countries. Bioactive compounds that may prevent and treat infectious diseases are identified based on their ability to inhibit bacterial neuraminidase (NA). We aimed to isolate and identify bioactive compounds from leaves and stems of *P. japonicas* (PJA) and elucidate their mechanisms of NA inhibition. Key bioactive compounds of PJA responsible for NA inhibition were isolated using column chromatography, their chemical structures revealed using ^1^ H NMR, ^13^ C NMR, DEPT, and HMBC, and identified to be bakkenolide B (**1**), bakkenolide D (**2**), 1,5-di-*O*-caffeoylquinic acid (**3**), and 5-*O*-caffeoylquinic acid (**4**). Of these, **3** exhibited the most potent NA inhibitory activity (IC_50_ = 2.3 ± 0.4 μM). Enzyme kinetic studies revealed that **3** and **4** were competitive inhibitors, whereas **2** exhibited non-competitive inhibition. Furthermore, a molecular docking simulation revealed the binding affinity of these compounds to NA and their mechanism of inhibition. Negative-binding energies indicated high proximity of these compounds to the active site and allosteric sites of NA. Therefore, PJA has the potential to be further developed as an antibacterial agent for use against diseases associated with NA.

## 1. Introduction

*N*-acetyl neuraminic acid (Neu5Ac), also known as sialic acid, is a major constituent of cell surface glycoproteins [1]. They are present in terminal positions of the sugar chain on the glycoconjugates, either alone or in oligo- or polymeric form. Sialic acid plays an important role in the regulation of cellular and molecular recognition in many biological events [2,3,4]. As a result of their exposed position, sialic acids are vulnerable to the action of sialidases. Sialidase (Enzyme entry: EC 3.2.1.18), also called neuraminidase (NA), belongs to the family of exo-glycosidases, and hydrolyzes terminal sialic acid from cell surface glycoproteins [5,6]. Viruses, microorganisms, and vertebrate synthesize NA. The enzyme has a role in lysosomal catabolism and the modulation of functional molecules involved in many biological processes. Disturbance of their expression and biosynthesis can lead to cell differentiation and cell adhesion in the inflammatory processes, malignant cell transformation, and pathogenesis of various diseases [7].

Bacterial pathogens can use NA for food scavenging, invasion, immune-suppression, and dissemination within the host. Recently, the bacterial NA was identified as a major virulence factor exacerbating sepsis [8]. Sepsis arises from a massive overreaction of the immune system to bacterial pathogens, leading to widespread edema and harm to vital tissues. Severe sepsis results in a high mortality rate in critically ill patients. The bacterial NA cleaves sialic acid from the glycoprotein ligand of a sialic acid-binding immunoglobulin lectin inhibitory receptor (SiglecG/10), abrogating its ability to dampen the immune response [9]. In some cases, they play an essential role in dissemination of infection and biofilm formation [10,11]. This indicates that bacterial NA could promote abnormal inflammatory responses leading to subsequent morbidity. Therefore, bacterial NA inhibitors have great potential as a therapeutic strategy against sepsis.

*Petasites japonicas* (Siebold & Zucc.) Maxim, known as giant butterbur, is a perennial plant belonging to the *Petasites* genus, and is a member of the Compositae/Asteraceae. It is widely distributed in Korea, Japan, and East China at higher altitudes [12,13]. It is traditionally used to treat furunculosis, contusion, wounds, and snakebites [14]. The young leaves of wild or cultivated species are consumed fresh as a vegetable, and are processed into different food products. Previous phytochemical investigations have shown that *Petasites* species contain many bioactive components, such as triterpenoids, sterols, fatty acids, phenolic compounds, sesquiterpenoids (e.g., bakkenolide), and other trace minerals [15,16,17,18,19]. Some of these compounds are responsible for the antioxidant, anti-allergenic, anti-inflammation, anti-carcinogenic, anti-mutagenic, antimicrobial, and neuroprotective effects of *Petasites* products [20,21,22,23,24]. To date, no report exists on bacterial NA inhibition by chemical constituents of *P. japonicus*. In this study, we have demonstrated the inhibitory effect of a methanol extract from leaves and stems of *P. japonicus* on bacterial NA and isolated active compounds from the extract based on the enzyme activity assay. Enzyme kinetic analyses and molecular-modeling studies have been performed using the most active compound to provide insights into the interactions between the active compounds and bacterial NA.

## 2. Material and Methods

### 2.1. General Experimental Procedures

CC was performed using Diaion HP-20 (Mitsubishi-Chemical, Tokyo, Japan), silica gel (230–400 mesh; Merck Co., Darmstadt, Germany) and Sephadex LH-20 (GE Healthcare Bio-Science AB, Uppsala, Sweden). Recycling Preparative HPLC was performed by using a LaboACE LC-5060 (JAI Co., Ltd., Tokyo, Japan). A JAIGEL-ODS column (20 × 500 mm, 15 μm, JAI) was used for preparative HPLC. Enzymatic assays were carried out on a Spectra Max M3 Multi-Mode Microplate Reader (Molecular Device, Sunnyvale, CA, USA). The NMR spectra of **1**–**4** were recorded on Bruker 700 and 900 MHz spectrometer (Bruker, Karlsruhe, Germany), using standard Bruker pulse programs. Chemical shifts are given as *δ*-values with reference to tetramethylsilane (TMS) as an internal standard. ^1^H and ^13^C-NMR assignments were determined by gHSQC, gHMBC, and ^1^H-^1^H-COSY. Neuraminidase (*C. welchii*), quercetin, 4-methylumbelliferyl-*N*-acetyl-α-d-neuraminic acid sodium salt hydrate (MU-Neu5Ac), and Tris buffer were purchased from Sigma-Aldrich (St. Louis, MO, USA). All other chemicals used were of biochemical reagent grad.

### 2.2. Sample Collection

The *P. japonicus* whole plants were collected in Baekdudaegan National Arboretum in April 2019 (voucher BDNA-2019-3004). Voucher specimen is deposited in the herbarium of Baekdudaegan National Arboretum for future reference, and the plant species were identified by the taxonomist Lee D.H. This plant was dried at room temperature in an airtight place under dark conditions.

### 2.3. Extraction and Isolation

The dried and powdered leaves and stems of *Petasites japonicus* (1.6 kg) were successively extracted using methanol, at room temperature (2 times at 3-day intervals, totaling 6 days). After the solvent was removed under reduced pressure at 45 °C, a residue (33.9 g) was obtained. After the removal of methanol under reduced pressure, the MeOH extracts was passed thought a Diaion HP-20 column and partitioned between *n*-hexane, chloroform, ethyl acetate, and *n*-butanol and water, in that order. Since the *n*-BuOH-soluble fraction showed enzyme inhibition against bacterial NA, this active fraction (11.4 g) was chromatographed over a silica gel column, with CHCl_3_:MeOH = 25:1 as the eluent, to give five fractions (PB1–PB5). PB3 (1.7 g) was separated by chromatography over Sephadex LH-20 with 95% methanol, to give five subfractions. Sub-fraction 2 was re-chromatographed on a Sephadex LH-20 column eluted with 85% aqueous methanol. The latter three subfactions were purified by preparative HPLC, using 70% aqueous methanol as the mobile phase to yield compound **1** (15 mg) and **2** (17 mg). PB5 (2.1 g) was subjected to a column of ODS (reversed-phase resin), eluting with a gradient of increasing methanol (20–100%) in water, and re-chromatographed on a column of Sephadex LH-20 with 70% aqueous methanol, to give three subfractions. The first subfraction was further separated by preparative HPLC with 75% aqueous methanol to give compound **3** (32 mg) and **4** (11 mg). All the isolated compounds were identified on the basis of spectroscopic data and comparisons with previous studies. Moreover, each compound achieves over 95% purity (Appendix A).

#### 2.3.1. Bakkenolide B (**1**)

^1^H-NMR (700 MHz, Chloroform-*d*) *δ* 5.91 (1H, dd, *J* = 7.2, 15 Hz, H-3′), 5.72 (1H, d, *J* = 11.2 Hz, H-9), 5.17 (1H, s, H-13a), 5.14 (1H, s, H-13b), 5.10 (1H, m, H-1), 4.63 (2H, m, H-12), 2.78 (1H, dd, *J* = 11.2, 5.0 Hz, H-10), 2.21 (1H, d, *J* = 14.3 Hz, H-6), 1.91 (1H, d, *J* = 14.3 Hz, H-6), 1.91 (3H, s, H-2′′), 1.85 (3H, dd, *J* = 7.2, 1.6 Hz, H-4′), 1.78 (2H, m, H-2), 1.75 (3H, s, H-5′), 1.66 (1H, m, H-3), 1.55 (1H, m, H-4), 1.34 (1H, m, H-3), 1.09 (3H, s, H-15), 0.87 (3H, d, *J* = 6.8 Hz, H-14). ^13^C-NMR (175 MHz, Chloroform-*d*) *δ* 177.5 (C-8), 169.9 (C-1′′), 167.3 (C-1′), 147.7 (C-11), 136.7 (C-3′), 128.2 (C-2′), 108.3 (C-13), 80.8 (C-9), 70.6 (C-12), 70.5 (C-1), 54.9 (C-7), 51.4 (C-10), 45.8 (C-6), 43.4 (C-5), 35.2 (C-4), 29.5 (C-3), 26.8 (C-2), 20.9 (C-2′′), 20.3 (C-5′), 19.5 (C-15), 15.5 (C-14), 15.5 (C-4′).

#### 2.3.2. Bakkenolide D (**2**)

^1^H-NMR (900 MHz, Chloroform-*d*) *δ* 7.04 (1H, d, *J* = 10.1 Hz, H-3′), 5.76 (1H, d, *J* = 11.2 Hz, H-9), 5.62 (1H, d, *J* = 10.14 Hz, H-2′), 5.21 (1H, s, H-13), 5.17 (1H, s, H-13), 5.15 (1H, m, H-1), 4.67 (2H, m, H-12), 2.75 (1H, dd, *J* = 11.2, 5.0 Hz, H-10), 2.39 (3H, s, H-4′), 2.24 (1H, d, *J* = 14.3 Hz, H-6), 2.02 (3H, s, H-2′′), 1.95 (1H, d, *J* = 14.3 Hz, H-6), 1.84 (1H, m, H-2), 1.76 (1H, m, H-2), 1.67 (1H, dd, *J* = 14.1, 3.6 Hz, H-3), 1.57 (1H, m, H-4), 1.37 (1H, dd, *J* = 12.9, 3.7 Hz), H-3), 1.11 (3H, s, H-15), 0.90 (3H, d, H-14). ^13^C-NMR (225 MHz, Chloroform-*d*) *δ* 177.5 (C-8), 169.9 (C-1′′), 165.6 (C-1′), 152.8 (C-3′), 147.8 (C-11), 112.4 (C-2′), 108.2 (C-13), 80.8 (C-9), 70.5 (C-12), 70.3 (C-1), 54.9 (C-7), 51.7 (C-10), 45.8 (C-6), 43.3 (C-5), 35.3 (C-4), 29.5 (C-3), 26.8 (C-2), 21.2 (C-2′′), 19.5 (C-15), 19.2 (C-4′), 15.5 (C-14).

#### 2.3.3. 1,5-di-*O*-Caffeoylquinic Acid (**3**)

^1^H-NMR (700 MHz, Methanol-*d*_4_) *δ* 7.64 (1H, d, *J* = 15.9 Hz, H-7′), 7.60 (1H, d, *J* = 15.9 Hz, H-7″), 7.08 (2H, s, H-2′,2″), 6.97 (2H, m, H-6′,6″), 6.81 (1H, s, H-5′), 6.79 (1H, s, H-5″), 6.38 (1H, d, *J* = 15.9 Hz, H-8′), 6.28 (1H, d, *J* = 15.9 Hz H-8″), 5.46 (1H, m, H-3), 5.42 (1H, m, H-5), 4.00 (1H, dd, *J* = 7.4, 2.8 Hz, H-4), 2.35 (1H, dd, *J* = 13.8, 3.0 Hz, H-6), 2.25 (2H, m, H-2), 2.19 (1H, m, H-6). ^13^C-NMR (175 MHz, Methanol-*d*_4_) *δ* 176.3 (C-7), 167.5 (C-9′), 167.0 (C-9″), 148.1 (C-4′), 148.0 (C-4″), 145.9 (C-7′), 145.7 (C-7″), 145.3 (C-3′,3″), 126.5 (C-1′), 126.4 (C-1″), 121.7 (C-6′,6″), 115.1 (C-5′,5″), 114.2 (C-8′), 113.9 (C-2′,2″), 113.7 (C-8″), 73.5 (C-1), 71.3 (C-3), 70.6 (C-5), 69.4 (C-4), 36.4 (C-2), 34.7 (C-6).

#### 2.3.4. 5-*O*-Caffeoylquinic Acid (**4**)

^1^H-NMR (900 MHz, Methanol-*d*_4_) *δ* 7.58 (1H, d, *J* = 15.9 Hz, H-7′), 7.07 (1H, d, *J* = 1.9 Hz, H-2′), 6.97 (2H, dd, *J* = 8.2, 1.8 Hz, H-6′), 6.80 (1H, d, *J* = 8.1 Hz, H-5′), 6.29 (1H, d, *J* = 15.9 Hz, H-8′), 5.36 (1H, m, H-5), 4.20 (1H, m, H-4), 3.75 (1H, dd, *J* = 8.6, 3.0 Hz, H-3), 2.18 (1H, m, H-6), 2.18 (1H, m, H-2), 2.08 (1H, m, H-2), 2.07 (1H, m, H-6). ^13^C-NMR (225 MHz, Methanol-*d*_4_) *δ* 175.7 (C-7), 167.2 (C-9′), 148.2 (C-4′), 145.6 (C-7′), 145.4 (C-3′), 126.4 (C-1′), 121.6 (C-6′), 115.1 (C-5′), 113.9 (C-8′), 113.8 (C-2′), 74.9 (C-1), 72.3 (C-4), 70.6 (C-3), 70.1 (C-5), 37.6 (C-2), 36.9 (C-6).

### 2.4. Bacterial Neuraminidase Inhibition Assay and Kinetics

The ability of compounds to inhibit the neuraminidase from *Clostridium perfringens* (EC 3.2.1.18) was evaluated with the following procedures and compared to quercetin. A previously reported method [25], with minor modifications, was used in performance of the NA inhibition assay. Substrate (MU-Neu5Ac) at the final 0.1 mM, was mixed with 90 μL of 50 mM Tris buffer (pH 7.5) at room temperature. Then, 10 μL of sample solution and 10 μL of NA (0.2 units/mL) were added to a well in a 96-well black immune-microplate (SPL life science, Pocheon, Korea). The mixture was recorded at excitation and emission wavelengths of 365 nm and 450 nm, with a Spectra Max M3 (Molecular Device, Sunnyval, CA, USA). Quercetin was used as a positive control with an IC_50_ value of 21.8 μM in this assay system. The 50% inhibitory concentration (IC_50_) for enzymatic activity of NA was determined from the dose-response curve. Each assay was conducted as three separated replicates.
Enzyme activity (%) = [1 − ([I]/IC_50_)] × 100(1)

Kinetic parameters were determined using the Lineweaver-Burk double-reciprocal-plot and Dixon plot methods at increasing concentrations of substrates and inhibitors. Inhibition constants (*K*_i_) were determine by Dixon plot. All parameters were then calculated using SigmaPlot (SPCC Inc., Chicago, IL, USA).

### 2.5. Molecular Docking

Crystal structure of neuraminidase from *Clostridium perfringens* (Protein Data Bank (PDB) entry, 5TSP) was obtained from RCSB (Research Collaboratory for Structural Bioinformatics) PDB (Protien Data Bank) for molecular docking study. The overall docking process was performed using diverse tools in Discovery Studio (DS) 2018 (BIOVIA, San Diego, CA, USA). Hydrogen atoms were added to the model and minimized to make a stable energy conformation, and to relax the conformation from close contacts. The three-dimensional (3D) structures of *N*-acetylneuramic acid (NANA), bakkenolide D (**2**), 1,5-di-*O*-caffeoylquinic acid (**3**, 1,5-DCOQ), and 5-*O*-caffeoylquinic acid (**4**, 5-COQ) were obtained from PubChem Compound in NCBI (National Center for Biotechnology Information). Ligands were prepared by Prepare Ligand protocol, for input into further protocol performing tasks, such as removing duplicates and enumerating isomers and tautomers. Ligands were docked into the active-site, which was defined by the site record contained in the PDB file, using the C-DOCKER module. Potential allosteric sites for bakkenolide D (**2**) were predicted by the Define and Edit Binding Site tool with the from Receptor Cavities option, with a radius of 5 Å or more opening site. The ligand orientation giving the lowest interaction energy was chosen for subsequent rounds of docking. An estimation of the binding energy between receptor and ligand (Δ*E*_Binding_) was calculated, using the equation *E*_Binding_ = *E*_Complex_ − *E*_Ligand_ − *E*_Receptor_.

### 2.6. Data Processing and Statistical Analysis

All the experiments were conducted in triplicate. The results were subjected to variance analysis using Sigma Plot (version 14.0, Systat Software, Inc., San Jose, CA, USA). Differences were considered significant at *p* < 0.05.

## 3. Results

### 3.1. Isolation of Inhibitors from P. japonicus

The methanol extracts of the aerial portions of *P. japonicus* showed the highest potency for enzymatic inhibitory activities against bacterial NA. The dried leaves and stems of *P*. *japonicus* were extracted with 95% MeOH, which was passed through a Diaion HP-20 column and solvent-partitioned successively to *n*-hexane, EtOAc, *n*-BuOH, and water extracts. The *n*-BuOH extract showed potent activity, and the extract was fractionated to identify the active compounds. Chemical analysis of these extracts using successive column chromatography over silica gel and Sephadex LH-20, and preparative HPLC resulted in the isolation and identification of four compounds. Their structures were elucidated as bakkenolide B (**1**), bakkenolide D (**2**), 1,5-di-*O*-caffeoylquinic acid (1,5-DCOQ) (**3**), and 5-O-caffeoylquinic acid (5-COQ) (**4**), using spectroscopic data with literature values (Figure 1) [22,26,27,28].

### 3.2. Bacterial Neuraminidase Inhibitory Activities

To evaluate the bacterial NA activity of isolated compounds from *P. japonicus*, NA from *Clostridium perfringens* was used (Table 1). The enzyme activity was assayed following a published protocol using the hydrolysis of 4-methylumbelliferyl-α-D-*N*-acetylneuraminic acid sodium salt hydrate. Quercetin was used as a positive control in the bacterial NA inhibition assays. All compounds tested decreased NA activity in a concentration-dependent manner. The compounds **2**, **3**, and **4** showed considerable activity against NA, with IC_50_ values of 2.3–80.1 μM, but compound **1** was inactive (IC_50_ > 200 μM, Figure 2a). Two caffeoylquinic acids (**3** and **4**) inhibited bacterial NA significantly, with IC_50_ values of 2.3 and 22.6 µM, respectively. They showed much better inhibition than the bakkenolide-type sesquiterpenes (**1** and **2**). These results showed that compound **3** exerts NA inhibitory activity that is more than 10 times higher than quercetin.

### 3.3. Kinetic Analysis of Inhibitors on NA

Three compounds showed potent NA inhibitory activity, so we further examined the inhibitory modes of these compound using Lineweaver-Burk plots of 1/velocity (1/*V*) versus 1/[S] and Dixon plots of 1/*V* maxapp versus concentration of compounds, and the most suitable mode was selected using the SigmaPlot 14.0 software (Figure 3 and Table 1). The inhibitory behaviors were found to be different according to the differences between the chemical structures of the two bakkenolides and two caffeoylquinic acids. Bakkenolide D (**2**) was estimated to be a non-competitive inhibitor, because increasing the substrate concentrations resulted in plots with the same *x*-Axis intercept as the uninhibited enzyme, but different slopes and *y*-intercepts (Figure 3a).

### 3.4. Molecular Docking Study of the Inhibition of NA by Inhibitors

Inhibitors such as bakkenolide D (**2**), 1,5-DCOQ (**3**), and 5-COQ (**4**) were docked on to the active, such as NA. The top scored docking pose was selected from the 50 poses generated with each ligand such as 1,5-DCOQ (**3**), 5-COQ (**4**), or the control substrate, *N*-acetylneuraminic acid (NANA). In the selected pose, NANA formed strong hydrogen bonds with residues R266, Q493, R555, Y587, R615, and Y655, π-donor hydrogen bond with F353, and charge interactions with residues R266, R555, and R615 in NA active site (Figure 4a). Similar to NANA, 5-COQ (**4**) formed hydrogen bonds and showed charge interactions with residues R266 and R615 and additionally formed hydrogen bonds with residues D328, W354, and Y485 in NA. A benzene ring of 5-COQ was found to interact with residues F327 and F353 by π-alkyl interaction and π-π stacked interaction, respectively (Figure 4b). 1,5-DCOQ (**3**) also interacted with residues R266, Q493, Y587, R555, and W354 by hydrogen bonds, similar to docked poses of NANA and 5-COQ (**4)**. Similar to 5-COQ, a benzene ring of 1,5-DCOQ was involved in a π-alkyl interaction with residues F327 in NA, while another benzene ring of 1,5-DCOQ was involved in a new π-cation interaction (Figure 4c). The binding energies (Δ*E*_Binding_) of NA with NANA, 1,5-DCOQ, and 5-COQ were −130.98 kcal mol^−1^, −152.66 kcal mol^−1^, and −181.85 kcal mol^−1^, respectively, indicating that enzyme-ligand complex is more stable in the order of binding with 5-COQ > 1,5-DCOQ > NANA, and this order is also consistent with the order of the *K*_i_ values. The higher stability is presumably owing to the hydrophobic interactions with the benzene rings of 1,5-DCOQ (**3**) and 5-COQ (**4**), not present in NANA. Bakkenolide D (**2**) did not dock to the active site, further corroborating the results obtained from the Lineweaver-Burk plots that bakkenolide D (**2**) could be a noncompetitive inhibitor. Docking of bakkenolide D (**2**) was attempted at nine predicted binding sites, because it could not be docked onto the active site (Appendix A). Among them, bakkenolide D (**2**) was only docked at site 1, 2, and 8, and the top 10 docking poses with high score were compared at each site (Appendix A). The docking poses at site 1 had low C-DOCKER interaction energy and binding energy values, suggesting that the poses at site 1 had highest reliability and stability. Thus, site 1 could be a putative allosteric binding site. At site 1, bakkenolide D (**2**) formed strong hydrogen bonds with residues K272 and R467, π-alkyl interactions with residues F271 and I332, and showed sulfur interaction with residues K472 in NA (Appendix A). Therefore, these compounds would be useful in the development of preventive and therapeutic agents.

## 4. Discussion

*P. japonicus* produces numerous compounds that have beneficial effects on human physiology. In particular, our study showed that *P. japonicus* extracts have the highest potency in inhibiting enzymatic activities of bacterial NA. The constituents of this plant that play such roles remain unknown; therefore, it was of interest to determine the bioactive compounds in *P. japonicus* extracts responsible for NA inhibition. The extraction yield and biological activity of the resulting extract are not only affected by number of extraction but also by the extraction solvent be rewritten. To prepare the extracts, air-dried samples were soaked in methanol at room temperature and were extracted two times at three-day intervals. To this end, four main compounds were isolated and identified from the extracts of the leaves and stems of *P. japonicus* for inhibitory effects against enzymatic activities of bacterial NA. The exact chemical structures of the four compounds (**1**, **2**, **3** and **4**) were fully determined from spectroscopic data including 2D-NMR (Appendix A). For example, the ^1^H-, ^13^C-NMR, ^1^H-^1^H COSY and HMBC spectra of the most effective compound (**3**) showed double caffeoyl substituted quinic acid derivatives. ^1^H-NMR spectra exhibited four doublets with coupling constant of 15.9 Hz, characteristic for trans olefinic protons (*δ*_H_ 7.64. 7.60, 6.38, 6.28). The coupling pattern of six aromatic proton signal (7.08 (2H), 6.97 (2 H, 6.81, 6.79 (each 1H)) appearing as two ABX systems in ^1^H-^1^H COSY spectrum, which indicated the presence two caffeic acid moieties. The rest of the signals of 1H-NMR were attributed to quinic acid moiety. The attachments of the two caffeoyl moieties at C-1 and C-5 of quinic acid were deduced from the HMBC correlation of H-1, with their ester carbonyl carbons (C-9′ and C-9″) at dc 167.0 (C-9″) and 167.5 (C-9′), respectively. These spectral data suggested that **3** was 1,5-dicaffeoylquinic acid. The identified compounds (**1**, **2**, **3** and **4**) were screened for their NA inhibitory activities at different concentrations. This forms the first intensive study showing that caffeoylquinic acids (**3** and **4**) are good NA inhibitors. Our results suggest that the capacity of these compounds to inhibit bacterial NA is affected by number of caffeoyl groups of the quinic acid moiety. For example, the 1,5-DCOQ **3** (IC_50_ = 2.3 ± 0.4 μM) was 11 times more effective than 5-COQ **4** (IC_50_ = 22.6 ± 0.6 μM). Additionally, the structural difference between compounds **1** and **2** were in the presence of the angeloyloxy group or 3-methylthioacryloyloxy group substituent on C-9, but the 3-methylthioacryloyloxy group was seven times more effective, compared with the angeloyloxy group. Thus, we found that the 3-methylthioacryloyloxy group of bakkenolide-type sesquiterpene plays a crucial role in bacterial NA inhibition. To the best of our knowledge, this study is the first report on the inhibitory activity of these compound’s (**2**, **3**, and **4**) on bacterial NA in the enzyme-based screening system. The enzyme-based screening suggested that this compound could bind with the enzyme on a site that is not an active site. The two caffeoylquinic acids (**3** and **4**) fell into the competitive inhibitor category. This competitive behavior was confirmed because *V*_max_ remained constant with the increasing concentration of the inhibitor (**3** and **4**), whereas *K*_m_ decreased (Figure 3b,c). This indicated that **3** and **4** can competitively interact with the substrate-binding site to inhibit enzyme activity when co-treated with various concentrations of substrates. Moreover, the plots of the remaining enzyme activity versus the concentrations of the enzyme at different inhibitor concentrations resulted in a group of straight lines, which all passed through the origin. These data suggested that the most potent compound **3** is a reversible inhibitor (Figure 2b). A Dixon plot is a well-accepted method for analyzing the enzyme inhibition type and determining the *K*_i_ value for inhibitor binding, wherein the *K*_i_ is represented by the value of the *x*-Axis. As shown representatively in Figure 3d–e and Table 1, the *K*_i_ values were 82.7 ± 0.4 μM, 1.4 ± 0.05 μM, and 16.1 ± 0.5 μM. The *K*_i_ values represent concentrations required to produce half-maximal inhibition; thus, inhibitors with lower *K*_i_ values may exhibit greater bacterial NA inhibition. These results support the observed interactions between the inhibitor molecules (**2**, **3**, and **4**) and NA.

## 5. Conclusions

In summary, we systematically investigated the inhibitory activity of the *n*-butanol fractionated chemical components of *P. japonicus* against bacterial NA. A successive column chromatography approach was used to separate the major compounds of the active fraction, including bakkenolide B (**1**), bakkenolide D (**2**), 1,5-di-*O*-caffeoylquinic acid (**3**), and 5-*O*-caffeoylquinic acid (**4**). Inhibition of bacterial NA by each compound was assessed by enzymatic evaluation and molecular docking. Caffeoylquinic acid compounds (**3** and **4**) were shown to be competitive inhibitors against bacterial NA, fitting into the active sites of NA in enzyme kinetic and molecular docking studies. However, bakkenolide D (**2**) was shown to be a noncompetitive inhibitor, because it could be docked at an allosteric site. Most importantly, caffeoylquinic acids emerged as potent inhibitors of bacterial NA, an important antibacterial drug target implicated in pathogenesis. Thus, *P. japonicus* could be an excellent source of bacterial NA inhibitors for food and medicinal uses.

## Figures and Tables

**Figure 1 biomolecules-10-00888-f001:**
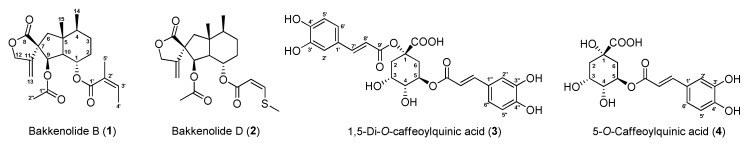
Chemical structures of isolated compounds from the leaves and stems of *P. japonica*.

**Figure 2 biomolecules-10-00888-f002:**
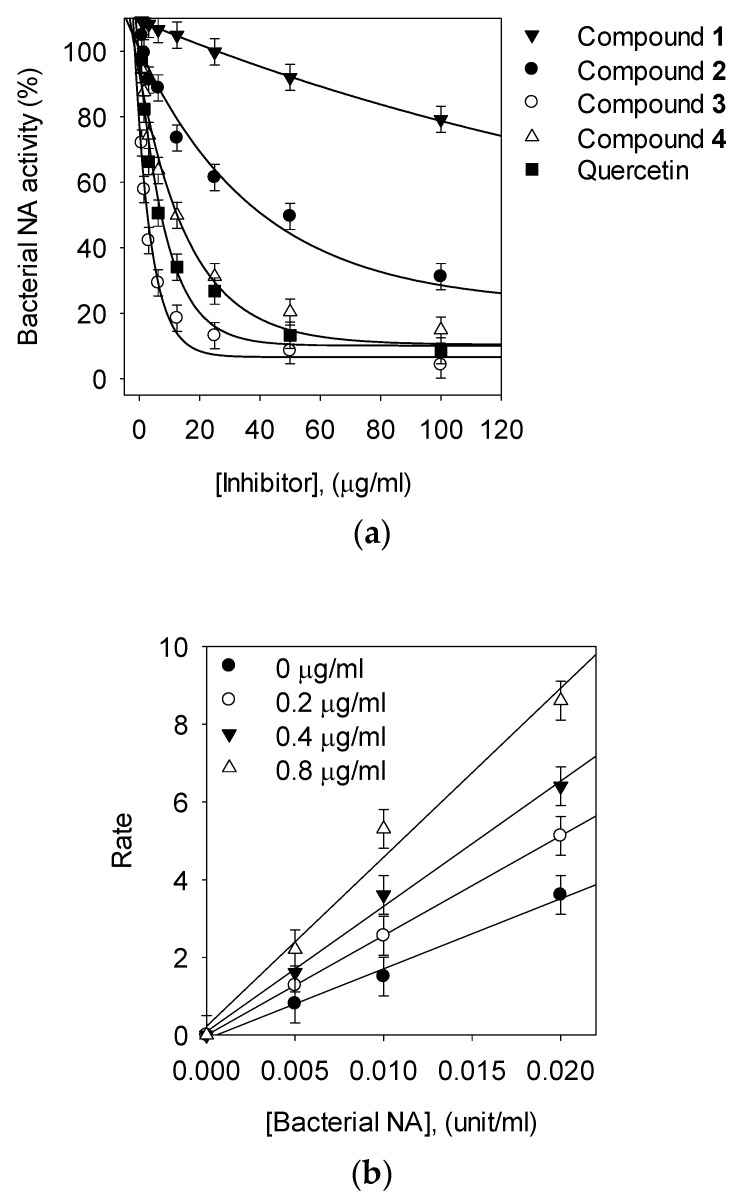
(**a**) Effects of compounds on bacterial neuraminidase (NA) catalyzed hydrolysis of 4-methylumbelliferyl-*N*-acetyl-α-d-neuraminic acid sodium salt hydrate (MU-Neu5Ac) (compound **1**, ▼; compound **2**, ●; compound **3**, ○; compound **4**, △; quercetin, ■; respectively). (**b**) Bacterial NA hydrolytic activity in the presence of compound **3** [0 μg/mL, ●; 0.2 μg/mL, ○; 0.4 μg/mL, ▼; 0.8 μg/mL, △].

**Figure 3 biomolecules-10-00888-f003:**
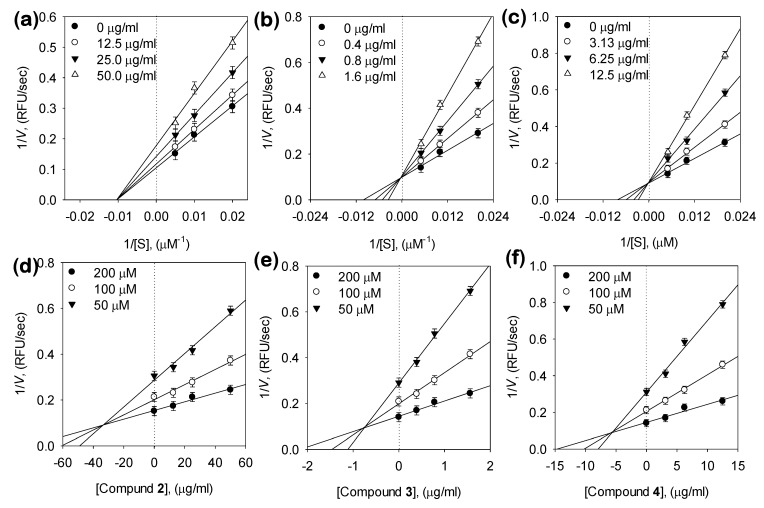
(**a**–**c**) Lineweaver-Burk plot for the inhibition of compounds (**2**, **3**, and **4**) on the hydrolytic activity of NA. Conditions were as followers: 0.1 mM MU-Neu5Ac, 50 mM Tris buffer (pH 7.5), at 38 °C, (**d**–**f**) Dixon plots for NA inhibition of compounds (**2**, **3**, and **4**), in the presence of different concentrations of substrate. The graphical symbols are substrate concentrations (50 μM, ▼; 100 μM, ○; 200 μM, ●).

**Figure 4 biomolecules-10-00888-f004:**
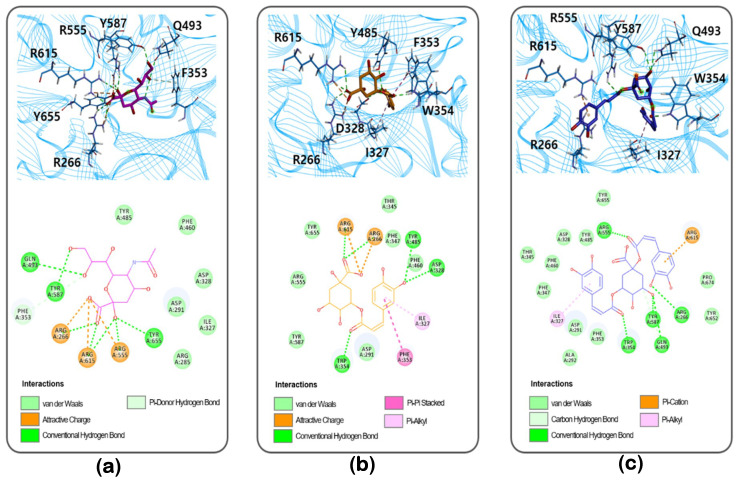
Docking poses of (**a**) *n*-acetylneuramic acid, (**b**) 5-*O*-caffeoylquinic acid, and (**c**) 1,5-di-*O*-caffeoylquinic acid. The 3D- and 2D-structures represent receptor-ligand interaction. *N*-Acetylneuramic acid, 5-*O*-caffeoylquinic acid, and 1,5-di-*O*-caffeoylquinic acid are represented as purple, orange, and blue stick models, respectively.

**Table 1 biomolecules-10-00888-t001:** Inhibitory effects of isolated compounds **1**–**4** on bacterial neuraminidase activities.

Compound	Bacterial Neuraminidase (*Clostridium Perfringerns*)		
	IC_50_ ^a^ (μM)	Type of Inhibition	*K*_i_^b^ (μM)
Bakkenolide B (**1**)	>200	NT ^c^	NT
Bakkenolide D (**2**)	80.1 ± 1.8	Noncompetitive	82.7 ± 0.4
1,5-di-*O*-caffeoylquinic acid (**3**, 1,5-COQ)	2.3 ± 0.4	Competitive	1.4 ± 0.05
5-*O*-Caffeoylquinic acid (**4**, 5-COQ)	22.6 ± 0.6	Competitive	16.1 ± 0.5
Quercetin	21.8 ± 0.7	NT	NT

^a^ The 50% inhibitory concentration (IC_50_) values (μM) were calculated from a log dose inhibition curve using bacterial neuraminidase as a substrate, respectively. ^b^ Values of inhibition constant. ^c^ NT is not tested.

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
