# Peer review of "Bakkenolides and Caffeoylquinic Acids from the Aerial Portion of Petasites japonicus and Their Bacterial Neuraminidase Inhibition Ability"

_biomolecules, 2020, doi:10.3390/biom10060888_

Round 1

Reviewer 1 Report

This manuscript described the isolation of four compounds from Petasites japonicus aerial part. Among these compounds, 1,5-di-O-caffeoylquinic acid has more potential inhibition for the bacterial neuraminidase. The enzyme kinetic parameters and molecular docking for neuraminidase also prove the competitive enzyme inhibition activity. However, because the four compounds were isolated by authors, the purity of compounds determined by HPLC or 1H-NMR spectra should be provided in material and methods section. In supplementary files these spectra seem contains other impurity. The coupling constants (J) of 1H-NMR data also need clear assign in L108-136. The NMR data needs descending rearrange according to the chemical shift (ppm) value. The detector for the recycling preparative HPLC should provide. How many cycles for purify the compounds 1 and 2? Authors can provide the HPLC fingerprint chromatogram of crude extract and fractions.    

Other minor points should be corrected to improve the quality of manuscript.

L74, Mitsuvishi.....sialica gel  -->Mitsubishi.....silica gel

L75, Sepahdex --> Sephadex 

L81, d-values --> δ-values

L138, Clostridium perfringens should be italic

L141,  Substrate.....D-neuramnidase acid....-->Substrate.....D-neuraminic acid....   (MU-Neu5Ac)

L141, the enzyme and substrate were purchase form what company?

L183 and all manuscript, 1,5-dicaffeoylquinic acid -->1,5-di-O-caffeoylquinic acid 

Figure 1, The sturcure of compounds 1 and 2 should give the position numbers as same as compounds 3 and 4. The CH3 should be CH3

Table 1, IC50 ( M) and Ki ( M)  --> μM

Table 1 legend, IC50 values (mM) --> (μM) 

Figure 4. (B) O-caffeoylquinic acid -->5-O-caffeoylquinic acid

L273, ....dc  --> .... δc

L262, L303 and others, P. japonicus should be italic

Author Response

For Reviewer 1

This manuscript described the isolation of four compounds from Petasites japonicus aerial part. Among these compounds, 1,5-di-O-caffeoylquinic acid has more potential inhibition for the bacterial neuraminidase. The enzyme kinetic parameters and molecular docking for neuraminidase also prove the competitive enzyme inhibition activity.

However, because the four compounds were isolated by authors, the purity of compounds determined by HPLC or 1H-NMR spectra should be provided in material and methods section. In supplementary files these spectra seem contains other impurity.

=> The purity of the four compounds was confirmed by high performance liquid chromatography (HPLC). The HPLC analyses shown in Figure S23 indicated that the purities of the separated compounds (1-4) all exceed 95%.

The coupling constants (J) of 1H-NMR data also need clear assign in L108-136. The NMR data needs descending rearrange according to the chemical shift (ppm) value.

=> We added and rearranged the coupling constants (J) and chemical shift (ppm) values suggested by the reviewer.

The detector for the recycling preparative HPLC should provide. How many cycles for purify the compounds 1 and 2?

=> We used recycling preparative HPLC only for preparative HPLC. As it may cause confusion among reviewers and subscribers, the recycling preparative HPLC was modified for the preparative HPLC. The results of preparative HPLC analysis were attached to Supplementary Materials.

Authors can provide the HPLC fingerprint chromatogram of crude extract and fractions.  

=> The HPLC analysis showed a retention time of approximately 12.1 min for bakkenolide D (2), 15.3 min for 1,5-di-O-caffeoylquinic acid (3), 16.3 min for bakkenolide B (1), and 17.2 min for 5-O-caffeoylquinic acid (4).

Other minor points should be corrected to improve the quality of manuscript.

L74, Mitsuvishi.....sialica gel  -->Mitsubishi.....silica gel

L75, Sepahdex --> Sephadex 

L81, d-values --> δ-values

L138, Clostridium perfringens should be italic

L141,  Substrate.....D-neuramnidase acid....-->Substrate.....D-neuraminic acid....   (MU-Neu5Ac)

L141, the enzyme and substrate were purchase form what company?

L183 and all manuscript, 1,5-dicaffeoylquinic acid -->1,5-di-O-caffeoylquinic acid 

Figure 1, The sturcure of compounds 1 and 2 should give the position numbers as same as compounds 3 and 4. The CH3 should be CH3

Table 1, IC50 ( M) and Ki ( M)  --> μM

Table 1 legend, IC50 values (mM) --> (μM) 

Figure 4. (B) O-caffeoylquinic acid -->5-O-caffeoylquinic acid

L273, ....dc  --> .... δc

L262, L303 and others, P. japonicus should be italic

=> In order to improve the quality of manuscript, we have revised all the lists pointed out by reviewers.

Reviewer 2 Report

This paper deals with the inhibitory effects of Petasites japonicas extracts on bacterial neuraminidase. The bioactive compounds responsible for this enzyme activity were isolated and identified by spectroscopic analysis. The topic is of interest and falls into the scope of the journal. In general, the paper is well-structured but concerns regarding treatment and methanolic extraction of the plant material should be further elucidated.

Plant bioactive compounds, especially phenolics, are susceptible to oxidation. Thus, long drying and long extraction procedures (7 days) favour oxidation processes with the loss of biological activity. Freeze-drying and the adoption of alternative fast-way extractions such as sonication, together with application of low temperature (4 °C), can avoid most oxidation and the action of lytic enzymes released during cell rupture. These aspects should be more detailed and considered in relation to the characterization of methanolic bioactive compounds.

Author Response

For Reviewer 2

Comments and Suggestions for Authors

This paper deals with the inhibitory effects of Petasites japonicus extracts on bacterial neuraminidase. The bioactive compounds responsible for this enzyme activity were isolated and identified by spectroscopic analysis. The topic is of interest and falls into the scope of the journal. In general, the paper is well-structured but concerns regarding treatment and methanolic extraction of the plant material should be further elucidated.

Plant bioactive compounds, especially phenolics, are susceptible to oxidation. Thus, long drying and long extraction procedures (7 days) favour oxidation processes with the loss of biological activity. Freeze-drying and the adoption of alternative fast-way extractions such as sonication, together with application of low temperature (4 °C), can avoid most oxidation and the action of lytic enzymes released during cell rupture. These aspects should be more detailed and considered in relation to the characterization of methanolic bioactive compounds.

=> Extraction is the crucial first step in the analysis of medicinal plants, because it is necessary to extracts the desired chemical components from the plant materials for further separation and characterization. The basic operation included steps, such as pre-washing, drying of plant materials or freeze drying, grinding to obtain a homogenous sample and often improving the kinetics of analytic extraction and also increasing the contact of sample surface with the solvent system. Proper actions must be taken to assure that potential active constituents of the extract from plant samples. As plants contain many compounds and may sometime act synergistically or antagonistically. Polar solvents are frequently used for recovering polyphenols from plant matrices. Ethanol has been known as a good solvent for polyphenol extraction and is safe for human consumption. Methanol has been generally found to be more efficient in extraction of lower molecular weight polyphenols, whereas aqueous acetone is good for extraction of higher molecular weight flavanols.

You can have a look at “Phytochemical methods a guide to modern techniques of plant analysis” by Jeffrey B. Harborne, different plants and different tissues give different concentration and also in varying proportion.

Round 2

Reviewer 1 Report

Authors has already improved the manuscript. However, there still has some minor point needs check. 

The 1H NMR data need more clear assign for all four compounds.  For example, the 1H NMR data of two H-6 at 2.21 and 19.1 ppm of bakkenolide B (1) should be doublet and the geminal coupling constant (J) is easy to calculate. For H-3", H-4", and H-10 at 5.91, 1.85, and  2.78 ppm the coupling pattern are doublet-doublet but no show J value?

For easy to read the structure of compounds 1 and 2 in Fig. 1 needs show the full structures with assign position number.    

Compound 4 is 5-O-caffeoylquinic acid but the structure showed 3-O-caffeoylquinic acid?

For compounds 3 and 4, R1 should in C1 position and R2 should be in C5 position.

In Fig. S23, the purity of compound 4 may less than 95% as authors declared. 

For 5-O-caffeoylquinic acid, the O should be italic.

In Fig. S21 and S22, please label the color zone with compounds 1, 2, 3, 4. The IR detector is typing error of RI detector?

Author Response

The 1H NMR data need more clear assign for all four compounds.  For example, the 1H NMR data of two H-6 at 2.21 and 19.1 ppm of bakkenolide B (1) should be doublet and the geminal coupling constant (J) is easy to calculate. For H-3", H-4", and H-10 at 5.91, 1.85, and 2.78 ppm the coupling pattern are doublet-doublet but no show J value?

=> We have revised the coupling constant of four compounds pointed out by the reviewer.

For easy to read the structure of compounds 1 and 2 in Fig. 1 needs show the full structures with assign position number.    

=> We revised Fig. 1 according to the reviewer’s comment.

Compound 4 is 5-O-caffeoylquinic acid but the structure showed 3-O-caffeoylquinic acid?For compounds 3 and 4, R1 should in C1 position and R2 should be in C5 position.

=> It is thought that some errors occurred in the design process of Figure 1. We identified the wrong part and corrected it.

In Fig. S23, the purity of compound 4 may less than 95% as authors declared. 

=>The purity of compound 4 may be less than 95 % in the HPLC analysis results, but a high purity results was obtained in the NMR analysis results. By combining the two results, the purity of the compound 4 was expressed as 95% or more.

For 5-O-caffeoylquinic acid, the O should be italic.

=>We corrected typographical errors to proper one.

In Fig. S21 and S22, please label the color zone with compounds 1, 2, 3, 4. The IR detector is typing error of RI detector?

=>The part pointed out by the reviewer was added, and ‘IR’ was corrected to ‘RI’.

Reviewer 2 Report

Authors did not satisfied my concerns regarding methanolic extraction of plant material. They assert and agreed that this is a very crucial step. However, they dis not detail in the text, as requested, how they avoided oxidation processes favoured by long drying and long extraction procedures (7 days). Why did not they use freeze-drying  or low operation temperature (4°C)? Which precautions have they taken?  Some aspects should be detailed and discussed in the manuscript.

Author Response

Authors did not satisfy my concerns regarding methanolic extraction of plant material. They assert and agreed that this is a very crucial step. However, they did not detail in the text, as requested, how they avoided oxidation processes favored by long drying and long extraction procedures (7 days). Why did not they use freeze-drying or low operation temperature (4°C)? Which precautions have they taken?  Some aspects should be detailed and discussed in the manuscript.

=> I agreed with the reviewer’s comment very much. The extraction of polyphenolic compounds mainly depends on the analytical techniques, the nature of the solvents used which many vary depending on the sources and their order of magnitudes. High temperature, long extraction times, and alkaline environment cause polyphenolic compounds degradation. Addition of reducing agents and the use of inert atmospheres are also employed for the protection of polyphenolic compounds during extraction. Thus, it is necessary to develop an optimum and appropriate method for the proper extraction, purification, and characterization of polyphenolic compounds to achieve higher accuracy in the results.

The aim of this study was to evaluate isolate and identify active compounds in leaves and stems extract of P. japonicus. Previous phytochemical investigations have shown that Petasites species contain many bioactive components such as triterpenoids, sterols, fatty acids, phenolic compounds, sesquiterpenoids (e.g., bakkenolide), and other trace minerals. The purpose of extraction is to separate the soluble plant metabolites, leaving behind the insoluble cellular marc (residue). The yield of extraction depends on many factors that affect the extraction rate, such as solvent type, extraction time, temperature, pH, solid: solvent ratio, number of extraction steps, as well as the composition and physical characteristics of the samples such as matrix and particle size. Solid-liquid extraction is one of the most commonly used and easiest methods of extraction for the purification of bioactive compounds from plant extracts. To increase the yield and rate of metabolite extraction may also be obtained by using an acidified organic solvent such as methanol and ethanol.

Solubility of the bioactive compounds mainly depends on the chemical nature and polarity of the compounds. So, we have extracted twice methanol at room temperature for 7 days. Methanol is the best solvent to extract metabolites since the processing cost is low with a high yield. Conventional extraction is typically carried out at temperatures ranging from 20 °C to 50 °C.

=> We added some more details of ‘factors affecting the extraction’ to the discussion part according to reviewer’s suggestions as follow.

 â¸¢To increase the yield of metabolites extraction was extracted from the leaves and stems of P. japonicus twice with methanol at room temperature for a week. ⸥

Round 3

Reviewer 2 Report

English of this last version needs to be revised. In particular, sentences at lines 19-22, 64-65, 89- 90, 262-263, 270-273 should be rewritten.

Moreover, what reported at lines 89-90 should be better clarified: is the time of extraction 3 days or seven days as previously reported in the first version and at line 263 of the present paper?

Author Response

English of this last version needs to be revised. In particular, sentences at lines 19-22, 64-65, 89- 90, 262-263, 270-273 should be rewritten.

=> We have revised the points that the reviewer pointed out in consideration of spelling, grammar and context.

Moreover, what reported at lines 89-90 should be better clarified: is the time of extraction 3 days or seven days as previously reported in the first version and at line 263 of the present paper?

=> In order to reduce the misunderstanding of our experiment process, we have changed that part.

  ⸢The dried and powdered leaves and stems of Petasites japonicus (1.6 kg) were successively extracted using methanol, at room temperature (2 times at 3-day intervals, totaling 6 days).⸥

=> The above procedure was performed within a week (7 days).
